# TEXT STYLE TRANSFER WITH CONFOUNDERS

## ABSTRACT

Existing methods for style transfer operate either with paired sentences or distributionally matched corpora which differ only in the desired style. In this paper, we relax this restriction and consider data sources with additional confounding differences, from which the desired style needs to be inferred. Specifically, we first learn an invariant style classifier that takes out nuisance variation, and then introduce an orthogonal classifier that highlights the confounding cues. The resulting pair of classifiers guide us to transfer text in the specified direction, creating sentences of the type not seen during training. Experiments show that using positive and negative review datasets from different categories, we can successfully transfer the sentiment without changing the category.[1]

## 1 INTRODUCTION

Despite advances in neural text generation (Brown et al., 2020), fine-grained control over generated outputs remains challenging. Indeed, the ability to easily transfer output style by altering attributes such as sentiment, formality, genre, and personal styles would make text generation tools more appealing (Hu et al., 2017; Shen et al., 2017; Rao & Tetreault, 2018; Xu et al., 2012).

Solving style transfer typically requires the method to be able to disentangle what should be transferred from orthogonal aspects of sentences that ought to be kept intact. This disentanglement problem can be largely avoided in simple supervised scenarios with access to parallel sentences differing only in style (e.g., sentiment transfer with parallel negative and positive sentences, Figure 1.a). Recent approaches address a more difficult version of the task by dispensing with parallel sentences (Shen et al., 2017). Nevertheless, they assume that the corpora differing in style remain distributionally matched in other ways (e.g., sentiment transfer with negative and positive reviews from the same category of products, Figure 1.b).

However, data available for training style transfer models is rarely distributionally matched, and often involves changes other than style as well (e.g., sentiment transfer where negative and positive reviews come from different product categories, Figure 1.c). This means that the desired style difference is no longer illustrated directly as the difference between the two corpora, making the task substantially more challenging. More subtly, the model is asked to generate sentences it has not seen during training, since the generated sentences in a new style should not also reproduce those confounding differences present in the training data.

Solving style transfer with confounding cues requires us to also learn what the desired style difference is. This can be facilitated by dividing the data into two groups of different styles, while the sets within each group illustrate variation we do not wish to alter. In the example in Figure 1.c, reviews are divided into two groups according to their sentiment. Within a group, each dataset corresponds to a different product category which should be preserved. Besides sentiment transfer, this setup easily generalizes to other style transfer tasks, e.g., dialectal transfer. In this case, the groups will be divided according to the dialects, and sets within each group will represent different speakers whose personal style should be preserved.

Our model builds on invariant risk minimization (Arjovsky et al., 2019) to infer the style as an invariant distinction across different datasets from the two groups. The resulting style classifier leaves complementary aspects of sentences to be controlled (preserved). We can illustrate aspects orthogonal to style with new sets of environments and learn another invariant classifier. Together,

---

[1]Our code will be publicly released after the review process.

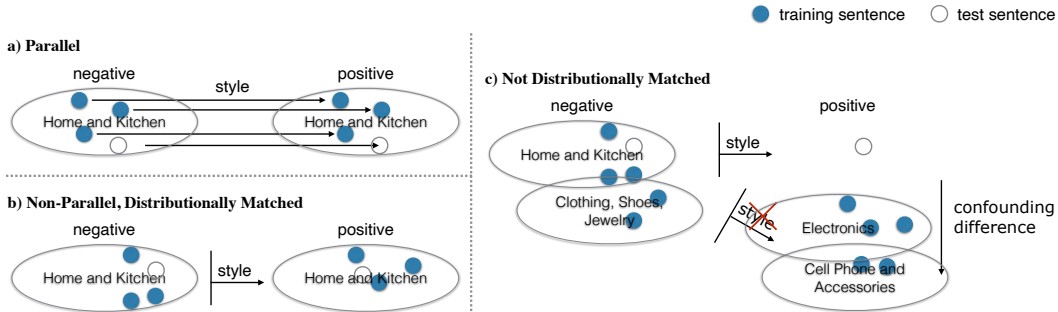

Figure 1: Different learning scenarios for style transfer. a) With parallel examples, learning of the transfer mapping is supervised. b) With non-parallel, distributionally matched datasets, learning of the transfer mapping is unsupervised. Nevertheless, style is given as the dataset difference, and generation takes place in-distribution. c) With non-parallel, not distributionally matched datasets, style needs to be inferred by excluding confounding differences. Not only is learning of the transfer mapping unsupervised, but generation is also out-of-distribution.

the two classifiers are used to guide sentence generation in a style transfer model along the desired direction. Combining with back-translation (Sennrich et al., 2015; Zhu et al., 2017) and language model regularization (He et al., 2020) techniques, we can generate new types of sentences that have not been seen during training.

We empirically evaluate our proposed model in two sentiment transfer settings that involve confounding factors. In the first setting, we augment original review data with special tokens, creating a spurious correlation with sentiment. In the second setting, we consider sentiment transfer where negative and positive reviews come from non-overlapping product categories. In both cases, we assess the ability of the model to transfer the sentiment, while preserving other aspects – special tokens in the first case, and product category in the second case. Our experiments demonstrate that our model successfully achieves both tasks, bringing significant gains over baselines that do not consider confounding factors. For instance, on the the task of sentiment transfer from different product categories, the model yields 28.4% increase in category preservation; according to human evaluation, its success rate is 6.2% higher than the best previous system.

## 2 RELATED WORK

The task of style transfer is related to paraphrasing, whose goal is to generate multiple linguistic realizations of the same underlying content (Androutsopoulos & Malakasiotis, 2010). However, style transfer adds an additional complexity – the requirement to control for a specific realization characteristic during rewriting. While paraphrasing models have been used in the past for the task of style transfer, these models are impacted by arbitrary variations present in paraphrasing datasets (Preotiuc-Pietro et al., 2016; Gröndahl & Asokan, 2019; Krishna et al., 2020). We instead take a data-driven approach for discerning style from content, where the style is not strictly limited to paraphrasing and can involve more general attribute transfer such as sentiment and political slant transfer (Prabhumoye et al., 2018).

Recent work in non-parallel style transfer proposes different techniques such as cross-alignment (Shen et al., 2017), delete and retrieval (Li et al., 2018), or parallel latent sequences (He et al., 2020). The generation of these approaches all takes place "in-distribution", i.e., realizing sentences of the type already seen during training. The sentences we generate are by design not seen during training (Figure 1.c). Subramanian et al. (2018) proposed multi-attribute style transfer that controls both the sentiment and the category of a sentence. However, similar to other prior work, they assumed access to all sentiment-category combinations for training. In contrast, we assume that negative and positive sentences come from non-overlapping categories.

Since we have to infer style from datasets with confounding cues, we make use of invariant learning (Peters et al., 2016; Arjovsky et al., 2019) to estimate style as an invariant direction. Moreover, our classifiers need to perform well across different combinations of source/target datasets and generalize

to generated text (out-of-distribution samples) to properly guide the transfer model. Therefore, our problem is also related to domain adaptation (Ben-David et al., 2010; Ganin & Lempitsky, 2015; Ganin et al., 2016) and domain generalization (Blanchard et al., 2011; Muandet et al., 2013). Our main contribution is the utilization of invariance for guiding the generation process.

## 3 STYLE TRANSFER WITH CONFOUNDERS

We consider two groups of datasets: $G_A = \{A_1, \ldots, A_n\}$ where each $A_i$ ($i = 1, \ldots, n$) is a dataset consisting of sentences with style $s_A$ (e.g., negative sentiment), and $G_B = \{B_{n+1}, \ldots, B_{n+m}\}$ where each $B_j$ ($j = n + 1, \ldots, n + m$) similarly conforms to style $s_B$ (e.g., positive sentiment). In addition to style, a dataset has its own characteristics different from each other (e.g., category). Our goal is to transfer a sentence $x$ of style $s_A$ into style $s_B$ (and vice versa) without changing its content or other characteristics.

One attempt is to aggregate collections of sentences in each group $A = A_1 \cup \cdots \cup A_n, B = B_{n+1} \cup \cdots \cup B_{n+m}$ and perform style transfer between them. However, the specific characteristics of $A_i$ and $B_j$ will become confounding factors and will be changed along with style. Instead, we notice that style is an invariant distinction between group $G_A$ and group $G_B$. In other words, the style difference is stable across different $A_i$ and $B_j$. Therefore, we can learn to isolate it by taking out intra-group variations. Once we have access to style, we can learn to transfer sentences along this direction while preserving other sentence characteristics.

We take a two-step procedure to accomplish this task. We first learn a pair of invariant classifiers to detect style and orthogonal characteristics. Then we use the classifiers to guide the learning of a style transfer model. Each part of the model is described in detail below.

### 3.1 INVARIANT CLASSIFIERS

We make use of Invariant Risk Minimization (Arjovsky et al., 2019, IRM) to learn our style and orthogonal classifiers. IRM requires us to specify a set of environments $\mathcal{E} = \{e_1, \ldots, e_E\}$, where each $e \in \mathcal{E}$ represents data $\{(x_i^e, y_i^e)\}_{i=1}^{n_e}$ collected under a certain environment. The different environments account for nuisance variation that the classifier should not pay attention to. The IRM objective is to learn a feature representation that enables a classifier to be simultaneously optimal for all environments. The rationale is that such representation likely involves primarily causal features that remain stable regardless of the nuisance variation. Therefore, the classifier can better generalize to new, unseen test environments compared to the standard Empirical Risk Minimization (ERM) classifier trained on the pooled data from all environments.

We adopt the IRMv1 formulation, which does not explicitly separate the representation from the classifier but treats the classifier output itself as the representation. In this vein, the objective becomes to minimize the classifier loss across all the data while penalizing per-environment gradients with respect to any multiplier of the classifier output:

$$\min_{\Phi:\mathcal{X}\to\mathcal{Y}} \sum_{e\in\mathcal{E}} R^e(\Phi) + \lambda\|\nabla_{w|w=1.0}R^e(w \cdot \Phi)\|^2 \tag{1}$$

where $R^e(f) := \mathbb{E}_{X^e,Y^e}[\ell(f(X^e), Y^e)]$ is the risk of $f$ under environment $e$ ($\ell$ can be any loss function), and $\lambda$ is a hyperparameter weighting the gradient penalty term. Gradients would be zero if $\Phi$ is per-environment optimal. It remains to define suitable environments for the invariant classifiers to separate style and confounding factors in our task.

### 3.1.1 STYLE CLASSIFIER

To learn a classifier to distinguish between styles $s_A$ and $s_B$ without relying the specific characteristics of each dataset, we pair each $A_i$ and each $B_j$ to form environments:

$$e_{i,j} = \{(x, y = 0) \mid x \in A_i\} \cup \{(x, y = 1) \mid x \in B_j\} \tag{2}$$

and learn an IRM classifier $C_s : \mathcal{X} \to \mathcal{Y}$ to predict the group label across environments $\{e_{i,j}\}_{1\leq i\leq n, n+1\leq j\leq n+m}$ (cf. Figure 2.a). Since all $A_i$ datasets share style $s_A$, and all $B_j$ datasets share style $s_B$, style is a feature representation that elicits an invariant classifier across different environments. Conversely, if the classifier uses any features specific to $A_i/B_j$, it will not be optimal in another environment consisting of a different pair $A_{i'}/B_{j'}$, thus violating the IRM constraint.

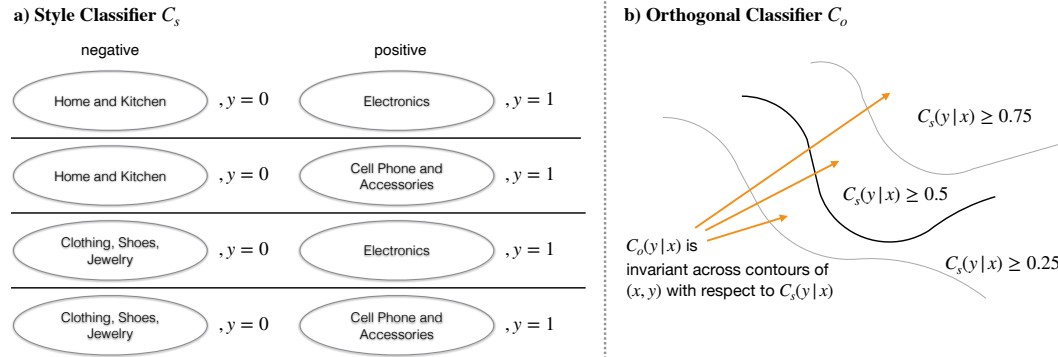

Figure 2: Illustration of the learning of the invariant classifiers. a) The style classifier $C_s$ is trained to be invariant across different pairs of $A_i$ and $B_j$ datasets. b) The orthogonal classifier $C_o$ is trained to highlight changes in sentences other than the style identified by $C_s$.

### 3.1.2 Orthogonal Classifier

In addition to requiring the transferred output to have a different style from the input, we also require it to retain the other characteristics of the input. To detect style-independent characteristics, we construct environments based on predictions of the style classifier $C_s$. These style-dependent environments serve to illustrate to the second invariant classifier $C_o$ what it should not rely on.

Let $D = \{(x, y = 0) \mid x \in A\} \cup \{(x, y = 1) \mid x \in B\}$ be the entire dataset, aggregate of all corpora. We create two environments[2]:

$$e_1 = \{(x, y) \in D \mid C_s(y|x) > 0.5\} \tag{3}$$
$$e_2 = \{(x, y) \in D \mid C_s(y|x) \leq 0.5\}$$

and learn an IRM classifier $C_o : \mathcal{X} \to \mathcal{Y}$ across $\{e_1, e_2\}$. In this way, $C_o$ cannot depend on the direction quantified by $C_s$ (i.e., the inferred style), and must find other orthogonal features to distinguish $A$ from $B$ (cf. Figure 2.b).

Note that if we have successfully transferred a sentence in $A_i$ to style $s_B$, then $C_s$ should predict label 1 for the output sentence because its style has been changed, while $C_o$ should continue to assign label 0 as the orthogonal characteristics ought to have remained intact.

### 3.2 Style Transfer Model

Based on the pair of invariant classifiers, we can now build a style transfer model to transfer a sentence to a different style specified by $C_s$ while preserving other characteristics controlled by $C_o$. Formally, we have dataset $D = \{(x, y)\}$, where $y$ denotes the group label of sentence $x$. We learn a style transfer model $M : \mathcal{X} \times \mathcal{Y} \to \mathcal{X}$ that takes a source sentence $x$ and a target group $y$ as input, and outputs a revised sentence that conforms to the style of group $y$. The model is learned first in a reconstruction phase and then in a transfer phase, discussed in turn.

When the input sentence $x$ is from group $y$, $M$ should behave as an autoencoder and reconstruct $x$. Therefore, we have the reconstruction loss:

$$\mathcal{L}_{\text{rec}}(\theta_M; x, y) = -\log p_M(x|x, y) \tag{4}$$

where $\theta_M$ denotes the parameters of $M$ to be learned. We first use $\mathcal{L}_{\text{rec}}$ to train $M$ on $D$ for an epoch to provide the model with a good initialization to generate realistic sentences.

After initializing $M$ as an autoencoder, we use the pair of invariant classifiers to guide it towards appropriate style transfer. Given an example $(x, y)$ in $D$, we let $\tilde{x} \sim p_M(\cdot|x, 1 - y)$ be the transferred output sampled from the model. If successfully transferred, $\tilde{x}$ should have style different from $x$ according to the style classifier $C_s$, and all the other characteristics should be the same as $x$ according

---

[2]A set of more segmented environments can be created based on the confidence of $C_s$. However, we did not observe performance gains with more environments.

to the orthogonal classifier $C_o$. Namely, $C_s$ should treat $\tilde{x}$ as coming from a different group, and $C_o$ should treat $\tilde{x}$ as coming from the same group. These two constraints lead to losses:

$$\mathcal{L}_{C_s}(\theta_M; y, \tilde{x}) = -\log p_{C_s}(1 - y|\tilde{x}) \tag{5}$$

$$\mathcal{L}_{C_o}(\theta_M; y, \tilde{x}) = -\log p_{C_o}(y|\tilde{x}) \tag{6}$$

We further use a pre-trained language model $L$ and introduce a KL divergence term $D_{\mathrm{KL}}(p_M(\cdot|x, 1 - y)\|p_L)$ to regularize the transfer distribution. Estimating the KL divergence with one sample of $\tilde{x} \sim p_M(\cdot|x, 1 - y)$, we have:

$$\mathcal{L}_{\mathrm{LM}}(\theta_M; x, y, \tilde{x}) = -\log p_L(\tilde{x}) + \log p_M(\tilde{x}|x, 1 - y) \tag{7}$$

The first term ensures the fluency of the generated sentence $\tilde{x}$. The second term corresponds to the negative entropy of the transfer distribution $-H_{p_M(\cdot|x, 1-y)}$. Maximizing this entropy term encourages exploration and helps to avoid bad local optima of constantly generating similar sentences (He et al., 2020).

Finally, we include the back-translation loss that the original sentence $x$ should be generated if we take $\tilde{x}$ as input and set the target label to be $y$ (Sennrich et al., 2015; Zhu et al., 2017):

$$\mathcal{L}_{\mathrm{BT}}(\theta_M; x, y, \tilde{x}) = -\log p_M(x|\tilde{x}, y) \tag{8}$$

Taken together, our overall training objective is:

$$\mathbb{E}_{(x,y)\sim D, \tilde{x}\sim p_M(\cdot|x, 1-y)}[\lambda_1 \mathcal{L}_{C_s} + \lambda_2 \mathcal{L}_{C_o} + \lambda_3 \mathcal{L}_{\mathrm{LM}} + \lambda_4 \mathcal{L}_{\mathrm{BT}}] \tag{9}$$

In practice, we use word-level loss for $\mathcal{L}_{\mathrm{LM}}$ and $\mathcal{L}_{\mathrm{BT}}$ (i.e., divided by the sentence length) to make them comparable in magnitude to $\mathcal{L}_{C_s}$ and $\mathcal{L}_{C_o}$. We use Gumbel-Softmax (Jang et al., 2016) to approximate the discrete sampling process of $\tilde{x}$ to compute gradients. At test time, we perform greedy decoding to generate the transferred output.

## 4 EXPERIMENTS

To assess the ability of our model to perform style transfer in the presence of confounders, we consider two experimental settings. In the first experiment, we compose a synthetic task by modifying sentence punctuation to create a spurious correlation with sentiment (Choe et al., 2020). The model needs to transfer the sentiment while preserving the punctuation. In the second experiment, we consider sentiment transfer across different product categories. The goal is to transfer sentiment without changing the product category.

**Baselines**   We consider two variants of our full model: the first is guided by the style classifier $C_s$ alone without the orthogonal classifier $C_o$; the second is guided by an ERM classifier $C_{\mathrm{ERM}}$ trained on $D$, performing direct transfer between $A$ and $B$ without taking into account confounders. We also compare with He et al. (2020), whose training objective has a back-translation loss and a KL divergence term similar to ours. They do not use a classifier, but rely on two language models separately trained on $A$ and $B$ to promote the transferred sentence to have the target style. Finally, we compare with a paraphrasing based method of Krishna et al. (2020), which again transfers between the aggregated $A$ and $B$ without considering confounding factors.

**Model Architecture**   We implement the classifiers $C_s, C_o$ using TextCNN (Kim, 2014) with sliding windows over 3-5 words. On the synthetic task, however, we found that because the dataset is simple to classify, the training loss of the CNN classifier is close to 0. This removes any guidance from IRM constraints, degenerating the resulting classifier to a standard ERM version. Therefore, to ensure that we estimate invariant classifiers, we use less powerful Bag-of-Words classifiers in the synthetic task. The employed language model $L$ is a 1-layer LSTM trained on $A \cup B$. The style transfer model $M$ consists of a 1-layer LSTM encoder and a 1-layer LSTM decoder augmented with attention mechanism (Bahdanau et al., 2014). To achieve style control through $y$, we learn two style embeddings for $y = 0, y = 1$ respectively, and add the corresponding one to each word embedding before feeding to the decoder.

Table 1: Classifier accuracy in the synthetic task when the spurious correlation is reversed.

| Model | Sentiment ACC | Punctuation ACC |
|---|---|---|
| $C_{\text{ERM}}$ | 54.4 | 45.6 |
| $C_s$ | **75.3** | - |
| $C_o$ | - | **89.9** |

**Training Regime and Hyperparameters**   During the training of the invariant classifiers, we linearly anneal the gradient penalty coefficient $\lambda$ from 0 to $\lambda_{\max}$ over the first 100K steps, and continue training with $\lambda = \lambda_{\max}$ for another 200K steps. We try $\lambda_{\max}$ in $\{1000, 3000, 5000\}$ and choose the one with the best performance on the validation set. In the training of the style transfer model $M$, we linearly anneal the temperature of Gumbel-Softmax from 1 to 0.1 in the first 20K steps, and then keep it at 0.1 in the next 10K steps. We try the weights of the loss terms $\{\lambda_1, \lambda_2, \lambda_3, \lambda_4\}$ in $\{1, 2, 4\}$, and select a model that strikes a good balance between output perplexity, BLEU with input, and accuracies according to the style and orthogonal classifiers. We note that different weight combinations will lead to different trade-offs (Pang, 2019), and we provide more detailed results in Appendix B.

### 4.1    SENTIMENT TRANSFER WITH DIFFERENT PUNCTUATION

We aim to emulate the presence of confounding factors by modifying sentences in a standard sentiment transfer dataset with special symbols. For a sentence $x$, we remove its original punctuation, and then add either an exclamation mark "!" or a period "." at the end. Let $p$ be the probability of adding "!", and $1 - p$ be the probability of adding ".". $p$ varies across different corpora. This setting enables us to measure model's ability to transfer sentence sentiment without changing its punctuation.

**Dataset**   We adapt the sentiment transfer dataset introduced by Shen et al. (2017), which has 177K negative sentences and 267K positive sentences for training, 2K negative and positive sentences for validation, and 500 negative and positive sentences for testing. We obtain new training, validation and test sets using the following procedure: (1) negative sentences are modified with $p = 0$ to form $A_1$; (2) positive sentences are equally divided into two sets, and modified with $p = 1, p = 0.8$ to form $B_2, B_3$ respectively. By construction, the punctuation strongly correlates with the sentiment. Therefore, a direct transfer between $A_1$ and $B_2 \cup B_3$ is likely to change the punctuation as well. By observing that sentiment is invariant while $p$ is variant in $B_2$ and $B_3$, we aim to transfer only the sentiment, not the punctuation.

**Evaluation**   We assess transfer accuracy by comparing change in sentiment in the input and output sentences. To automate this evaluation, we utilize a separate sentiment classifier trained on negative and positive sentences both modified with $p = 0.5$. In this way, its prediction will not be affected by punctuation. Note that this classifier is only used for evaluation, not for training the style transfer model. We also evaluate model's ability to keep punctuation intact during transfer by directly comparing input and output punctuation. To measure fluency, we report the perplexity of the output measured by an unbiased language model trained on sentences modified with $p = 0.5$. Finally, we compute the BLEU score of the output with respect to a human reference (Li et al., 2018).

**Results**   We first report performance of the invariant classifiers $C_s$ and $C_o$ in the setting reflecting their intended use in our style transfer model. We reverse the correlation between punctuation and sentiment labels to simulate style transferred sentences, and assess their ability to predict the desired aspects. Specifically, we test the accuracy of classifiers on positive sentences modified with $p = 0$ and negative sentences modified with $p = 1, p = 0.8$. Table 1 shows results of $C_s$ and $C_o$ compared with the standard $C_{\text{ERM}}$ classifier trained on $D$. As expected, $C_{\text{ERM}}$ performs poorly, mixing sentiment and punctuation clues. In contrast, the invariant classifier successfully separating the two aspects, achieving accuracy of 75.3% on sentiment prediction and 89.9% on punctuation prediction.

Table 2 summarizes style transfer results. The paraphrasing-based method of Krishna et al. (2020) is not suitable for sentiment transfer that requires semantic changes, as shown by its low sentiment accuracy of $28.4\%$. Moreover, it has the lowest BLEU score because it introduces unnecessary rewriting learned from the paraphrasing dataset. Despite reaching high sentiment accuracy, the direct

Table 2: Automatic evaluation results of the synthetic sentiment transfer task. Accuracies less than 30 are marked in red.

| Model | Sentiment ACC | Punctuation ACC | PPL | BLEU$_{ref}$ |
|---|---|---|---|---|
| Krishna et al. (2020) | 28.4 | 44.7 | 53.2 | 10.1 |
| He et al. (2020) | 82.2 | 4.6 | **27.0** | 20.4 |
| $M$ w/ $C_{ERM}$ | 65.1 | 4.5 | 40.4 | **28.8** |
| $M$ w/ $C_s$ | 70.4 | 5.5 | 43.3 | 27.2 |
| $M$ w/ $C_s, C_o$ (Ours) | **84.3** | **97.7** | 48.1 | 24.3 |
| Input Copy | 2.4 | 100.0 | 34.0 | 32.8 |
| Reference | 76.8 | 100.0 | 42.3 | 100.0 |

Table 3: Example outputs of the synthetic sentiment transfer task.

| | |
|---|---|
| Input | the sales people here are terrible . |
| Reference | the sales people are great . |
| Krishna et al. | the people here are absolutely terrible . |
| He et al. | the sales people here are great ! |
| $M$ w/ $C_{ERM}$ | the sales people here are amazing ! |
| $M$ w/ $C_s$ | the sales people here are fantastic ! |
| Ours | the sales people here are amazing . |

| | |
|---|---|
| Input | great food but horrible staff and very very rude workers . |
| Reference | great food and excellent staff and very very nice workers . |
| Krishna et al. | great food , but very poor service . |
| He et al. | great food but excellent staff and very very friendly workers ! |
| $M$ w/ $C_{ERM}$ | great food and great staff and very very nice workers ! |
| $M$ w/ $C_s$ | great food but great staff and very very friendly workers ! |
| Ours | great food and great staff and very very friendly workers . |

| | |
|---|---|
| Input | the food is delicious and plentiful ! |
| Reference | the food was tough and dry ! |
| Krishna et al. | the food is delicious and plentiful ! ” . |
| He et al. | the food is mediocre and plentiful . |
| $M$ w/ $C_{ERM}$ | the food was mediocre and plentiful . |
| $M$ w/ $C_s$ | the food is mediocre and plentiful . |
| Ours | the food was mediocre , too ! |

| | |
|---|---|
| Input | excellent combination of flavors , very unique ! |
| Reference | the flavors are nothing to write home about ! |
| Krishna et al. | very unique combination of flavors , very unique ! ” . |
| He et al. | horrible customer service . |
| $M$ w/ $C_{ERM}$ | terrible combination of flavors , very disappointing . |
| $M$ w/ $C_s$ | terrible combination of flavors , not unique . |
| Ours | terrible combination of flavors , not outstanding ! |

Table 4: Automatic evaluation results of sentiment transfer from different categories.

| Model | Sentiment ACC | Category ACC | PPL | BLEU$_{src}$ |
|---|---|---|---|---|
| Krishna et al. (2020) | 22.6 | **57.4** | **35.8** | 19.2 |
| He et al. (2020) | 77.7 | 22.5 | 44.6 | 47.6 |
| $M$ w/ $C_{ERM}$ | **89.6** | 14.6 | 42.4 | 47.0 |
| $M$ w/ $C_s$ | 78.0 | 36.4 | 45.1 | **59.2** |
| $M$ w/ $C_s, C_o$ (Ours) | 79.9 | 50.9 | 49.2 | 57.4 |
| Input Copy | 3.1 | 75.3 | 34.5 | 100.0 |

transfer methods of He et al. (2020) and $M$ guided by $C_{ERM}$ have low punctuation accuracy ($4.6\%$ and $4.5\%$, respectively). Our full model achieves both high sentiment accuracy ($84.3\%$) and high punctuation accuracy ($97.7\%$). The reliance on the orthogonal classifier $C_o$ is proved critical – its omission results in dramatic drop of punctuation accuracy ($5.5\%$). Output examples in Table 3 provide multiple illustrations of these phenomena: Krishna et al. (2020) often paraphrases the input without changing the sentiment, while other models consistently change punctuation; only our full model successfully transfers the sentiment without changing the punctuation.

## 4.2 SENTIMENT TRANSFER WITH DIFFERENT CATEGORIES

Our second experiment focuses on sentiment transfer with product category as a confounding factor. Previous work took negative and positive reviews from the same category (Li et al., 2018). Our setting is more challenging where negative and positive reviews belong to distinct, non-overlapping categories. The model needs to infer that it is the sentiment to be transferred, not the category.

**Dataset** We use the 5-core Amazon review data (Ni et al., 2019), focusing on four large product categories: Clothing Shoes and Jewelry (CSJ), Home and Kitchen (HK), Electronics (E), and Cell Phone and Accessories (CPA). We further filter reviews based on their length, keeping reviews of length 5-20 words. Following standard practice, reviews with rating above three are considered positive, and those below three are considered negative. Reviews with rating three are discarded. Specifically, $A_1$ consists of negative reviews from the CSJ category, $A_2$ - negative reviews from HK, $B_3$ - positive reviews from E, and $B_4$ - positive from CPA. We create a dataset of 150K sentences for each category, in which 130K are used for training, 10K for validation and 10K for testing. Note that we do not use any review data of different sentiments from the same category.

**Evaluations** To assess sentiment transfer accuracy automatically, we train a classifier on complete data which includes negative and positive reviews from all four categories. It has test accuracy of $96.0\%$. Moreover, we train a 4-way category classifier using complete data to assess whether the output preserves input product category. It has test accuracy[3] $71.0\%$. As in the previous task, we report the perplexity of the output measured by an unbiased language model trained on the complete data. Note that these omniscient models are used only for evaluation and not for training the style transfer model. Since we do not have human references for this task, we compute the BLEU score with respect to source sentences as a rough indicator of content preservation (Zhao et al., 2018). In addition, we conduct human evaluation by asking human judges to rate each output on a Likert scale from 1 to 5 on three criteria: sentiment transfer, content preservation, and fluency. We consider a generated output successful if it is rated 4 or 5 on all three criteria. We evaluated 200 randomly sampled examples (100 negative and 100 positive) and collected two annotations for each sentence.

**Results** Table 4 shows automatic evaluation results for different models. Our model yields significant performance gains over the baselines, achieving $57.3\%$ absolute increase in sentiment transfer compared to Krishna et al. (2020), and $28.4\%$ increase in category preservation compared to He et al. (2020). The human evaluation results in Table 5 further verify the superiority of our model. Krishna et al. (2020) has the highest score in fluency as their model is fine-tuned from GPT2-Large. Our model achieves the highest sentiment transfer score and the highest success rate.

---

[3]This accuracy can be attributed to the fact that some short sentences do not have clear indicators of their product category.

Table 5: Human evaluation results of sentiment transfer from different categories.

| Model | Sentiment | Content | Fluency | Success |
|-------|-----------|---------|---------|---------|
| Krishna et al. (2020) | 1.9 | **4.2** | **4.4** | 11.8% |
| He et al. (2020) | 3.3 | 3.6 | 4.0 | 24.3% |
| Ours | **3.4** | 4.1 | 4.1 | **30.5%** |

Table 6: Example outputs of sentiment transfer from different categories.

| **Clothing, Shoes and Jewelry (negative → positive)** | |
|---|---|
| Input | this shirt was too tight . the sizing seems off . |
| Krishna et al. | the shirt is too tight . |
| He et al. | this case was great . the protection seems great . |
| Ours | this shirt works just perfect . the sizing seems well . |

| **Home and Kitchen (negative → positive)** | |
|---|---|
| Input | the containers do not lock well and are made of low quality materials . |
| Krishna et al. | the containers do not fit securely and are made from poor quality material . |
| He et al. | the phones work well and has made of sound quality of low quality materials . |
| Ours | the containers does the job well and are made of high quality materials . |

| **Electronics (positive → negative)** | |
|---|---|
| Input | exactly as advertised . converted a molex plug into a sata |
| Krishna et al. | the molex plug was convert to sata as advertised . |
| He et al. | way too big . leaves a inaccurate cut into a bath |
| Ours | not as advertised . converted a molex plug into a sata |

| **Cell Phones and Accessories (positive → negative)** | |
|---|---|
| Input | very sturdy and helpful to use while driving . |
| Krishna et al. | drives very well and is very useful . |
| He et al. | very stiff and weak to use while washing . |
| Ours | very thin and uncomfortable to use while driving . |

Example outputs in Table 6 show that our model successfully isolates sentiment from product category, preserving the latter during transfer. In contrast, He et al. (2020) mixes up categories by changing product specific nouns, such as rewriting "shirt" to "case", "containers" to "phones", and "driving" to "washing". Table 10 in the appendix provide more examples, some of which illustrate several failure modes of the model. For instance, when transferring sentiment of the sentence "very poor quality . crooked on one end .", the model only modifies the sentiment of the first clause, leaving the second clause intact. Another failure case is the use of inappropriate adjectives, such as rewriting "the drive works as designed" to "the drive is too large".

## 5 CONCLUSION

In this paper, we consider a new, challenging version of the style transfer task, where the model has to exclude confounders and infer the desired transfer direction from data. We propose to learn a pair of invariant classifiers to detect style and orthogonal characteristics, and then use them to guide a style transfer model. We empirically demonstrate significant performance gains over direct style transfer in two experimental settings. While this technology shows significant promise, it still requires further development to be used in practice. However, as style transfer algorithms continue to advance, their ability to support rapid, at scale content modification will increase. This can potentially result in dissemination of fake news and other forms of misinformation.

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

# APPENDIX

## A CLASSIFIER ACCURACY IN THE REAL TASK

Here, we report performance of the invariant classifiers $C_s$ and $C_o$ to classify based on sentiment and category respectively when their coupling is reversed (Section 4.2). We take positive reviews from CSJ and HK, and take negative reviews from E and CPA. The test accuracy in shown in Table 7.

Table 7: Classifier accuracy in the real task when sentiment-category coupling is reversed.

| Model | Sentiment ACC | Category ACC |
|-------|---------------|--------------|
| $C_{\text{ERM}}$ | 64.0 | 36.0 |
| $C_s$ | **80.4** | - |
| $C_o$ | - | **62.9** |

## B TRADE-OFF OF DIFFERENT WEIGHT COMBINATIONS OF LOSS TERMS

Table 8: Automatic evaluation results of sentiment transfer from different categories. Accuracies less than 30 are marked in red.

| Model | Sentiment ACC | Category ACC | PPL | BLEU$_{\text{src}}$ |
|-------|---------------|--------------|-----|---------------------|
| Krishna et al. (2020) | 22.6 | **57.4** | **35.8** | 19.2 |
| He et al. (2020) | 77.7 | 22.5 | 44.6 | 47.6 |
| $\lambda_{C_{\text{ERM}}}, \lambda_{\text{LM}}, \lambda_{\text{BT}} = 1, 1, 1$ | 87.8 | 15.8 | 42.8 | 48.0 |
| $\lambda_{C_{\text{ERM}}}, \lambda_{\text{LM}}, \lambda_{\text{BT}} = 2, 1, 1$ | 89.6 | 14.6 | 42.4 | 47.0 |
| $\lambda_{C_{\text{ERM}}}, \lambda_{\text{LM}}, \lambda_{\text{BT}} = 4, 1, 1$ | **90.2** | 10.3 | 49.2 | 47.7 |
| $\lambda_{C_s}, \lambda_{\text{LM}}, \lambda_{\text{BT}} = 1, 1, 1$ | 72.8 | 38.7 | 45.3 | 61.7 |
| $\lambda_{C_s}, \lambda_{\text{LM}}, \lambda_{\text{BT}} = 2, 1, 1$ | 78.0 | 36.4 | 45.1 | 59.2 |
| $\lambda_{C_s}, \lambda_{\text{LM}}, \lambda_{\text{BT}} = 4, 1, 1$ | 85.0 | 29.5 | 42.5 | 53.0 |
| $\lambda_{C_s}, \lambda_{C_o}, \lambda_{\text{LM}}, \lambda_{\text{BT}} = 1, 1, 1, 1$ | 68.9 | 56.6 | 49.8 | **63.4** |
| $\lambda_{C_s}, \lambda_{C_o}, \lambda_{\text{LM}}, \lambda_{\text{BT}} = 2, 1, 1, 1$ | 73.1 | 53.3 | 51.7 | 61.4 |
| $\lambda_{C_s}, \lambda_{C_o}, \lambda_{\text{LM}}, \lambda_{\text{BT}} = 4, 1, 1, 1$ | 79.9 | 50.9 | 49.2 | 57.4 |
| Input Copy | 3.1 | 75.3 | 34.5 | 100.0 |

## C  ADDITIONAL EXAMPLES

Table 9: Additional example outputs of the synthetic sentiment transfer task.

| | |
|---|---|
| Input | the new management team is horrible . |
| Reference | the new management team is great . |
| Krishna et al. | the new management team is terrible . |
| He et al. | the new management team is excellent ! |
| $M$ w/ $C_{\text{ERM}}$ | the new management team is amazing ! |
| $M$ w/ $C_s$ | the new management team is great ! |
| Ours | the new management team is great . |
| Input | did n't even get a response . |
| Reference | even got a response . |
| Krishna et al. | they didn't even respond to my question . |
| He et al. | did a great job and the staff is very friendly ! |
| $M$ w/ $C_{\text{ERM}}$ | did n't even get a response ! |
| $M$ w/ $C_s$ | did and even get a response ! |
| Ours | always great and always a response . |
| Input | i got my car detailed here and it looked amazing ! |
| Reference | car was detailed and came out bad ! |
| Krishna et al. | i had a car detailed and it looked incredible ! ) . |
| He et al. | i got my car detailed here and it looked awful . |
| $M$ w/ $C_{\text{ERM}}$ | i got my car detailed here and it looked terrible . |
| $M$ w/ $C_s$ | i got my car detailed here and it looked horrible . |
| Ours | i got my car detailed here , it looked awful ! |
| Input | they are super friendly , very helpful , and amazingly quick ! |
| Reference | so slow and rude , not helpful at all ! |
| Krishna et al. | they were very friendly , very helpful , and unbelievably quick ! ) . |
| He et al. | they are super rude , very helpful , and amazingly slow . |
| $M$ w/ $C_{\text{ERM}}$ | they are super rude , very unprofessional , and amazingly quick . |
| $M$ w/ $C_s$ | they are extremely rude , not helpful , and amazingly quick . |
| Ours | they were incredibly rude , not helpful , but amazingly quick ! |

Table 10: Additional example outputs of sentiment transfer from different categories.

| **Clothing, Shoes and Jewelry (negative → positive)** |
|---|

| | |
|---|---|
| Input | very thin material . and too short for me . i 'm 5'10 " |
| Krishna et al. | love it . just too thin , too short , and too short . |
| He et al. | very nice case . and great for iphone 's . i 'm 5'10 " |
| Ours | very nice product . and fits fine for me . i 'm 5'10 " |

| | |
|---|---|
| Input | too big for my face |
| Krishna et al. | too big for my face |
| He et al. | great fit for my s6 |
| Ours | works great for my face |

| **Home and Kitchen (negative → positive)** |
|---|

| | |
|---|---|
| Input | does not solve the lid problem ! |
| Krishna et al. | no way to fix the problem with the lid ! |
| He et al. | does the trick for everything problem ! |
| Ours | does the job the lid problem ! |

| | |
|---|---|
| Input | very poor quality . crooked on one end . |
| Krishna et al. | it is poor quality . |
| He et al. | very good quality . accurate on one end . |
| Ours | very good quality . crooked on one end . |

| **Electronics (positive → negative)** |
|---|

| | |
|---|---|
| Input | this is great . it easy to use |
| Krishna et al. | easy to use . |
| He et al. | this is too small . it had to return |
| Ours | this was broken . it difficult to use |

| | |
|---|---|
| Input | the drive works as designed |
| Krishna et al. | the drive works as designed |
| He et al. | the sized did not work properly |
| Ours | the drive is too large |

| **Cell Phones and Accessories (positive → negative)** |
|---|

| | |
|---|---|
| Input | it is durable and not as bulky as i thought |
| Krishna et al. | not as heavy duty as i thought it would be . |
| He et al. | it is uncomfortable and not as bulky as i thought |
| Ours | it is cheap and not as protective as i thought |

| | |
|---|---|
| Input | excellent quality and perfect fit |
| Krishna et al. | great quality and perfect fit |
| He et al. | horrible quality and returned it |
| Ours | extremely small and tight fit |

