# OpenReview forum: "Text Style Transfer with Confounders"
_ICLR.cc/2022/Conference — ICLR 2022 Submitted_

### Official Review · Reviewer_fSZh · 2021-10-30

**Correctness:** 4
**Technical Novelty And Significance:** 4
**Empirical Novelty And Significance:** 3
**Recommendation:** 6
**Confidence:** 4

**Main Review:**


**Strengths**

1. Style transfer is an important problem in natural language generation, with several practical applications (as discussed in [1]). The authors have identified a worrying issue in current style transfer systems (modification of non-target features like content), and attempted to fix this issue via invariant risk minimization (IRM).

2. The methods developed in this paper are quite novel --- to the best of my knowledge this is the first work using IRM for style transfer. IRM is a good fit for the confounder issue in style transfer, and I found the ideas in the paper interesting. The authors use an interesting set of loss functions to use these orthogonal classifiers to guide the model to perform style transfer without modification of confounders.

3. The authors test the approach on sentiment transfer, studying one synthetic confounder and one real-world confounder (product category). Experiments confirm the models trained by the model modify sentiment while keeping the confounder consistent. The authors test their method using both automatic and human evaluations, comparing it against strong baselines. On both kinds of metrics the authors see improvements using their proposed approach.

**Weaknesses**

1. My biggest concern with IRM is the need to identify environments across which the style classifiers should be consistent. In the sentiment transfer case, product category is certainly a valid confound which leads to semantic modification, but there could be several others, for instance intra-category variation of features of different products, writing style of author (formality / simplicity / usage of analogies). Obtaining these group annotations might be difficult in practice. The current approach also needs an orthogonal classifier, which needs a well-defined label space for the confounding factor. Many confounds related to semantics might be hard to precisely define & identify.

2. It would be useful to see the applicability of this approach on one more task which is not sentiment transfer. One possibility could be using the GYAFC formality transfer corpus, which has two categories of data (Entertainment & Music, Family & Relationships). Also while maybe this is out of the scope of this work, it will be interesting to see if confounding factors can be identified automatically. In a task like converting Shakesepare <---> Tweets (https://arxiv.org/abs/2010.05700), there will be several confounding features between the two styles. For instance, Shakespeare data will be on topics from the 1500s, whereas Tweets will mostly contain modern 21st century topics. These features will need to be disentangled from style (lexical / syntactic choice) for successful semantic preserving style transfer. Here it will be harder to segregate the corpus into suitable environments.

3. The ablation studies in Appendix B could be stronger --- what happens when you completely remove certain loss functions (weight = 0)? Also it's hard to compare between rows in this table since it's missing an "overall score" / aggregation like you did for the human evaluation.

**Minor**

1. The human evaluation is missing checking the accuracy of the confounding factor like you did for automatic evaluation ("CATEGORY ACC")

2. I would avoid branding this paper as "style transfer" since style generally refers to properties unrelated to semantics unlike sentiment (which is the focus of this work). The term "attribute transfer" might be more appropriate.

**Summary Of The Paper:**

Style transfer is a text generation task where a certain attribute of a sentence (like formality) is modified while preserving sentence content. It is generally studied in "unsupervised" settings --- no access to parallel data of sentences differing only in the target attribute. Nevertheless, algorithms assume access to corpora of unpaired sentences in each of the attributes, which may significantly differ in content. Prior works [1, 2, 3, 5] have noted this difference in content, which is encouraging models to modify non-target attributes like semantics (See Table 1 in [4]).

This paper is an attempt to fix this issue using invariant risk minimization (IRM) [6]. IRM encourages classifiers to be invariant across "environments", or datasets which share a target class but differ in other aspects which are not essential for classification ("spurious correlations"). In this work two spurious correlations are studied in sentiment transfer --- (1) a synthetic punctuation correlation; (2) product category of Amazon reviews. To perform style transfer, the authors first train two orthogonal IRM classifiers --- the first classifies the target attribute ("sentiment") irrespective of the confounder ("product category"), and the second for the confounder ("product category") irrespective of sentiment. These two classifiers are used to guide model generation, along with an LM smoothing and backtranslation objective (similar to [4]).

The authors compare their proposed approach against strong baselines and report promising results.

[1] - https://arxiv.org/abs/1910.03747
[2] - https://arxiv.org/abs/2010.05700
[3] - https://dl.acm.org/doi/10.5555/3016100.3016326
[4] - https://arxiv.org/abs/1811.00552
[5] - https://arxiv.org/abs/1905.13464
[6] - https://arxiv.org/abs/1907.02893

**Summary Of The Review:**

Overall, I thought the paper was well-written, had very interesting ideas and a good evaluation / comparison against prior work. Weakness #1/#2 will prevent me from going to the next higher score (of 8), but I'm in favour of acceptance.

---

> ### Author Response · Authors · 2021-11-22
> **Response to reviewer fSZh**
>
> We thank the reviewer for the feedback. We address each question in turn:
>
> **How to get group annotations**
>
> Our approach assumes that it is relatively easy to obtain group annotations, i.e., separating $A$ into $A_1,\cdots,A_n$ and $B$ into $B_{n+1},\cdots,B_{n+m}$ that highlight variability orthogonal to the style. In our experiments, we made use of the
> product category that was readily available. In addition, we could use for instance authorship labels or simply cluster the sets based on generic (irrelevant to style) attributes. We believe that our need for group annotations is more realistic than the strong assumption of existing methods that $A$ and $B$ have the same product and author distribution. We acknowledge, of course, that there are many choices and methods for obtaining group annotations, and we hope that future research will further investigate methodology for collecting datasets illustrating the desired style to be transferred and the unwanted confounders to be excluded. Besides, the problem of automatically designing environments in IRM has also been studied to some degree (Creager et al., 2021).
>
> We agree with the reviewer that many confounds can be hard to precisely define. Therefore, we believe that illustrating them via data and learning them through a classifier can be a practical way, which does not necessarily require a label space clearly defined to us.
>
> **More datasets**
>
> We agree with the reviewer that experimenting on more datasets will help to better understand the performance of the models. That said, we believe the datasets we used have shown limitations of existing methods and demonstrated the effectiveness of our proposed model to exclude confounders in style transfer. We will explore more datasets in the revision.
> (Our method requires that $n+m\ge 3$ to illustrate the confounder beyond the style, thus direct use of the GYAFC dataset with two categories is not suitable here.)
>
> For converting Shakespeare's text to tweets, indeed as the reviewer said, their topics are from different times which will confound with the style, and we only have $n=1$ for the 16th century and $m=1$ for the 21st century and cannot further segregate the corpus.
> Nevertheless, if we consider converting Shakespeare's sonnets to tweets, we can collect sonnets of different authors at different times (to get $n>1$), and this will illustrate that time and authorship are confounding factors and preserve them during transfer.
>
> **Ablation studies**
>
> The language model loss ensures the fluency of the generated sentence, and the back translation loss promotes content preservation --- completely removing them will cause significant performance degradation in the perplexity and the BLEU score respectively (detailed results are in "Ablation studies" in the response to reviewer 9yNQ).
>
> **Aggregation of automatic metrics**
>
> Unlike the human evaluation that uses a Likert scale of 1 to 5 in all aspects, different automatic metrics have different units and are therefore hard to aggregate. In particular, for the accuracies, it should be as high as possible; for the perplexity, it should be close to that of the input; for the BLEU score, it should be relatively high but not necessarily the higher the better. Moreover, the accuracies and the BLEU score are bounded (0-100), whereas the perplexity is unbounded.
>
> **Confounder assessment in human evaluation**
>
> The content preservation criterion in human evaluation includes preserving the confounding aspect (category). We'll clarify this in the revision.
>
> **References**
>
> Creager et al. (2021) Environment Inference for Invariant Learning

---

> > ### Comment · Reviewer_fSZh · 2021-11-24
> > **Thank you for your response**
> >
> > Thanks for your detailed response, I appreciate it.
> >
> > > Therefore, we believe that illustrating them via data and learning them through a classifier can be a practical way, which does not necessarily require a label space clearly defined to us.
> >
> > Could you explain this part more? Don't you need product category / punctuation labels (a "well-defined label space") for training the IRM classifiers?
> >
> > > (Our method requires that m+n=2 to illustrate the confounder beyond the style, thus direct use of the GYAFC dataset with two categories is not suitable here.)
> >
> > Isn't `m+n=4` for the GYAFC corpus? For each of the formal/informal styles, you have two domains (Family & Relationships as well as Entertaiment and Music) making `m=n=2` (see Table 2, 3 here - https://arxiv.org/pdf/1803.06535.pdf). If I understand correctly this is similar to your punctuation/sentiment setting.
> >
> > > Our approach assumes that it is relatively easy to obtain group annotations ... We agree with the reviewer that many confounds can be hard to precisely define ...... cannot further segregate the corpus
> >
> > Yes agreed, that's exactly my main worry with the current method. There are likely several confounds which you need to identify when dealing with data from the wild which will likely not be segregated into well-defined groups. Nevertheless, I agree the proposed method will be a great setup when a confound can be easily identified and the corpus can be segregated according to it.
> >
> > > We believe that our need for group annotations is more realistic than the strong assumption of existing methods that A and B have the same product and author distribution.
> >
> > Minor point, but I don't really think many prior methods assume this in their methodology, but I agree they end up transferring the confounding features, just like your results table shows. (Lample et al. 2018, He et al. 2020 try to extract semantics using backtranslation at training. Krishna et al. 2020 rely on a pretrained paraphrase generation model, and show results on style transfer between completely different content distributions like Shakespeare/Tweets).

---

> > > ### Author Response · Authors · 2021-11-28
> > > **Thank you for your reply!**
> > >
> > > **A well-defined label space for the confounding factor**
> > >
> > > Yes, we need labels for the group annotations. What we meant is slightly different. For example, consider transferring a Shakespeare's sonnet into a Shakespearean tweet, it can be difficult to define which of its linguistic features belong to Shakespeare's style that should be retained, and which belong to the sonnet style that should be transferred.
> > > Nevertheless, we think that we can learn the general sonnet style by inferring the invariance of sonnets of different authors.
> > > The authorship label here is an indirect supervision for the underlying general sonnet style not specific to Shakespeare, which may not be clearly defined on the surface (it may be difficult for humans to distinguish between the Shakespearean elements and the sonnet elements in a Shakespeare's sonnet).
> > >
> > > **GYAFC corpus**
> > >
> > > If we use the same categories for both styles, then category is no longer a confounding difference, and one can apply methods from previous work. Indeed, we could mix the datasets in different proportions to make category a confounding factor.
> > > However, our goal is to demonstrate that in the most challenging case where two styles have non-overlapping categories, we can still transfer the style without changing the category and create style-category combinations not seen during training.
> > >
> > > **Multiple confounders**
> > >
> > > Actually, in our setup, the datasets within a group can differ in multiple aspects, as long as they share the desired style. We think our approach can be generalized to multiple confounders.
> > > It would be interesting to explore this in future work.
> > >
> > > **Assumptions made by previous work**
> > >
> > > He et al. (2020) treats $A$ and $B$ as partially observed parallel data and assumes that the transferred $A$ has the same distribution as $B$, and vice versa (see their Fig. 1).
> > > In other words, they assume that $A$ and $B$ only differ in the desired style and are distributionally matched otherwise.
> > > Based on this, they hypothesize a parallel latent sequence for each observed sequence and derive ELBO, which consists of a back-translation loss and a KL divergence between the transferred (posterior) distribution and the (prior) distribution over the target domain (see their Eq. 3).
> > > The back-translation loss alone does not prevent changes in the confounding aspect --- one can change the style as well as the confounder and change them back in back translation.
> > > In fact, their KL divergence term promotes the transferred $A$ sentences to look like $B$, which in our setup will change both the style and the confounding factor.
> > >
> > > Krishna et al. (2020) first trains a model $M$ on a separate paraphrasing dataset, and uses it to paraphrase style $A$ sentences to $A'$ and similarly $B$ to $B'$.
> > > Then, they train inverse models $M_A$ to map $A'$ to $A$ and $M_B$ to map $B'$ to $B$. At test time, to transfer a sentence to style $A$/$B$, they first apply $M$ to paraphrase it and then apply $M_A$/$M_B$ to map to the target style. This approach relies on two assumptions:
> > >
> > > 1. The paraphrasing dataset covers the desired style transformation. Otherwise $M_A$ and $M_B$ are applied out-of-distribution. Consider sentiment transfer, for example, the paraphraser $M$ maintains the sentiment, and $M_A$ is trained to map negatives to negatives. During testing, however, $M_A$ will instead receive a positive sentence as input and needs to transfer it to negative, which results in low sentiment accuracy.
> > >
> > > 2. The paraphrasing dataset does not involve unwanted changes. Otherwise, when applying the paraphraser $M$, the sentence will correspondingly undergo unwanted changes. This is why this approach has low BLEU scores in our experiments, because the paraphraser often induces unnecessary changes in sentence structure.
> > >
> > > Our method does not rely on a paraphrasing dataset, which on the one hand allows us to transfer styles that are not strictly limited to paraphrasing, and on the other hand avoids being affected by arbitrary variations in the paraphrasing dataset that are irrelevant to the desired style.

---

> > > > ### Comment · Reviewer_fSZh · 2021-11-28
> > > > **Thanks for your reply, I will keep my score at 6**
> > > >
> > > > Thank you for the clarifications and detailed reply. I will keep my score of "marginally above an accept threshold" or a 6 since the weaknesses (#1, #2) I pointed out still hold true. Nevertheless, I think the ideas in the paper are very interesting and hence I gave it an accept score.
> > > >
> > > > > Indeed, we could mix the datasets in different proportions to make category a confounding factor. However, our goal is to demonstrate that in the most challenging case where two styles have non-overlapping categories
> > > >
> > > > Experiments on GYAFC (or one more dataset) will make this paper stronger, where you have category labels available. You could artificially create a fully non-overlapping setup as well --- taking formal sentences only from Family & Relationships and informal sentences from Entertainment and Music, for instance.

---

### Official Review · Reviewer_Lb6o · 2021-11-02

**Correctness:** 3
**Technical Novelty And Significance:** 2
**Empirical Novelty And Significance:** 2
**Recommendation:** 5
**Confidence:** 4

**Main Review:**

Pros:

+ interesting application and synthetic dataset

Cons:

- outdated encoder and text generator

- less general framework than [Subramanian et al., 2018]

==

I don't understand the synthetic noise addition by punctuation change: what are the changes?

Why relying on a so old model for text modeling? What would be the performance of BERT or T5 on the same task?
Gao, Z., Feng, A., Song, X., & Wu, X. (2019). Target-dependent sentiment classification with BERT. IEEE Access, 7, 154290-154299.

This framework is very reductive compared to [Subramanian et al., 2018] in which feeling is simply one attribute among others. One wonders what this approach would look like in the context of the experiments conducted in [Subramanian et al., 2018].

The authors propose to compare their approach with baselines that do not take into account all hypothesis... But they do not compare their work with existing approaches from the litterature.

Is it reasonnable to use a 1-layer LSTM for text generation purpose today? Even if GPT is very expensive, BART or T5 could have been investigated for such a task.

Regarding table 2, I understand that Puntuation ACC can be low but I don't get how Sentiment ACC can be so low whereas it is a binary classification task.

Generally speaking, sentiment accuracy seems not competitive with state of the art models, even if I am not familiar with the dataset from  Shen et al. (2017).

Comparing authors approach with Krishna et al. (2020) seems not relevant (even according to the authors)

**Summary Of The Paper:**

This article deals with style transfer. It proposes an approach to generate texts with opposite sentiments in a adversarial context where the style is correlated with other contextual factors such as domain or noisy punctuation.
Given the fact that other contextual factors are weakly labeled, the authors introduce Invariant Classifers to extract the style from other attributes.
The general framework has been defined by (Arjovsky et al., 2019, IRM) and the basic classifier is (Kim, 2014). Section 3.2 describes the authors' contribution where multiple orthogonal classifiers enables them to transfer style while preserving other attributes.
The authors propose relevant baselines corresponding to hypotheses ablations. The text generator is a 1-layer LSTM.
The authors demonstrate that their approach is able to transfer sentiment while preserving domain or punctuation depending on the experiment.

**Summary Of The Review:**

This work is interesting and consistant but the fact that both the encoder and the generator are outdated seems critical regarding a pure NLP paper. Despite some clear qualities, I have to reject this contribution.

---

> ### Author Response · Authors · 2021-11-17
> **Response to reviewer Lb6o**
>
> We thank the reviewer for the feedback. We respectfully disagree with some of reviewer’s criticisms. We hope that through our clarifications below, the reviewer can reevaluate our contribution. We will improve clarity in the revision, and we are always available for further discussions.
>
> **Use of LSTM & literature**
>
> The two baselines of He et al. (2020) and Krishna et al. (2020) that we compared against in the paper are state-of-the-art text style transfer models, where He et al. (2020) uses LSTM and Krishna et al. (2020) uses pretrained GPT-2. While we agree with the reviewer that using large pretrained models could help improve performance, we'd like to point out that LSTM is still actively used in the field (Prabhumoye et al., 2019; He et al., 2020; Yang and Klein, 2021). We chose to use LSTM as our method involves back-propagation through the model's generations, and LSTM is more efficient to train due to its fast decoding speed (d'Autume et al., 2019). Moreover, LSTM is competitive on relatively short text.
>
> **Comparison with Subramanian et al. (2018)**
>
> Subramanian et al. (2018) requires both negative *and* positive sentences for all categories in order to perform sentiment transfer without changing the category. In contrast, we only assume either negative *or* positive sentences for each category, so our task is more challenging than Subramanian et al. (2018). For applications such as dialectal transfer, most speakers speak one dialect, and different dialects have a different set of speakers. Our method can be used to transfer the dialectal style without changing the speaker’s personal style, whereas the method of Subramanian et al. (2018) is not applicable here.
>
> **Punctuation change**
>
> For negative sentences, we change their punctuation to ".". For positive sentences, we divide them into two sets of equal size.
> In the first set, the punctuation of all sentences is changed to "!"; in the second set, the punctuation of 80% of the sentences is changed to "!", and the remaining 20% is changed to ".". In this way, "." strongly correlates with negative sentiment, and "!" strongly correlates with positive sentiment.
>
> **Sentiment ACC**
>
> We'd like to clarify that our task is to generate sentences with the opposite sentiment while preserving other aspects, which is not just a binary classification. On the original dataset of Shen et al. (2017), the state-of-the-art results by He et al. (2020) have sentiment accuracy 87.90. We achieve sentiment accuracy 84.3 in the presence of an additional punctuation confounder.
>
> **References**
>
> He et al. (2020) A probabilistic formulation of unsupervised text style transfer
>
> Krishna et al. (2020) Reformulating unsupervised style transfer as
> paraphrase generation
>
> Prabhumoye et al. (2019) Towards Content Transfer through Grounded Text Generation
>
> Yang and Klein (2021). FUDGE: Controlled Text Generation With Future Discriminators
>
> d'Autume et al. (2019) Training Language GANs from Scratch
>
> Subramanian et al. (2018) Multiple-Attribute Text Style Transfer
>
> Shen et al. (2017) Style Transfer from Non-Parallel Text by Cross-Alignment

---

> > ### Comment · Reviewer_Lb6o · 2021-11-29
> > **Thank you for your responses & references**
> >
> > Thank you for this clear answer. I understand you argumentation but I am still embarrassed by the lack of range of experiences: several multi-aspect datasets exist and would enable you to validate your framework on a wider range than sentiment.
> > I get the argument on simple LSTM... But once again, I am convinced that it works only on a limited scope whereas pre-trained LM would probably allow for better performances in different situations.
> > I will keep my orginal mark.

---

### Official Review · Reviewer_6xK3 · 2021-11-02

**Correctness:** 2
**Technical Novelty And Significance:** 2
**Empirical Novelty And Significance:** 2
**Recommendation:** 3
**Confidence:** 4

**Main Review:**

Strengths

- The idea of using Invariant Risk Minimization is interesting.

Comments/questions on the proposed methods and evaluation

- In section 3.1.2, how to learn `an IRM classifier`? Or, to be specific, how this classifier can be implemented?
- In table 2, if the sentiment classifier (84.3) can outperform human reference (76.8), I am not sure whether we should trust it or how we should interpret its results?
- In section 4.2, it is a good idea of using BLEU with source sentences as a way to measure content preservation, however, this is not sufficient --- when input copy can give a perfect BLEU score, then we should not read too much from 59.2. Additional evaluation methods are needed here.
- Overall, the proposed method seems to be similar to relevant to the idea of learning discriminators for generation tasks, which has been used in prior work of text style transfer. Based on the description, I am not sure whether the proposed method can be better than learning a collection of discriminators, where each discriminator by design will tell or not tell the difference within an environment.
- As a method that has the potentials to be used in many other problems (as described in the introduction), it is disappointing to see both evaluation tasks are sentiment transfer. Would it be possible to at least try some other text style transfer tasks?


Comments on writing

- It would be great if the explanation of IRM in section 3.1 can be self-contained and based on the proposed task (text style transfer)
- In section 3.1.1, it says
> style is a feature representation that elicits an invariant classifier across different environment

In general, I don't think this statement is valid. By constructing the `environments`, it can eliminate other factors, but there is no guarantee that style is the only factor left.

**Summary Of The Paper:**

This paper presents a method that can work on a single text style transfer task with multiple datasets. One example scenario as exemplified in the paper is the sentiment transfer task with datasets on multiple product categories. The basic idea of this work is to design a cross alignment between datasets from different categories, so the learning algorithm can focus on the expected stylistic information and eliminate other confounding factors. The proposed methods were evaluated on the sentiment transfer task with two different setups to demonstrate the value.

**Summary Of The Review:**

The major concerns of this work are the novelty of the proposed method and the sufficiency of evaluation.

---

> ### Author Response · Authors · 2021-11-17
> **Response to reviewer 6xK3**
>
> We thank the reviewer for the feedback. We address each question in turn:
>
> **How to learn an IRM classifier**
>
> We learn an IRM classifier with SGD. At each training step, we take a minibatch of examples **$\{(x_i^e,y_i^e)\}_{i=1}^{m}$** from each environment $e$ and compute the loss $R^e(\Phi) + \lambda \|\nabla_{w|w=1.0} R^e(w\cdot\Phi) \|^2$ for classifier $\Phi$. The first term $R^e(\Phi)$ is the normal ERM loss $\frac{1}{m}\sum_{i=1}^m -\log p_{\Phi}(y_i^e|x_i^e)$. The second term is the squared norm of the gradient of the loss $R^e(w\cdot\Phi)$ with respect to $w$ after multiplying $\Phi$'s output logits with a fixed scalar $w=1.0$. The total loss is the sum of the losses of each environment. A PyTorch implementation can be found in Appendix D of Arjovsky et al. (2019).
>
> **Sentiment accuracy of human reference**
>
> We note that human reference does not have the highest accuracy as measured by the classifier. This is because in cases like "the man did not stop her", human reference is "the man stopped her", which reverses the source sentence but does not have a clear sentiment. The models will transfer the sentence to "the man loved it", thus passing the evaluator classifier.
>
> **BLEU with source sentences & evaluations**
>
> As the reviewer pointed out, a higher BLEU score with the source sentence is not necessarily better.
> Previous work uses it as a rough indicator of content preservation (Zhao et al., 2018; He et al., 2020).
> In addition to the BLEU score, we have also measured sentiment accuracy, category accuracy, and perplexity. By considering these metrics in full, we can have a comprehensive assessment of the performance of the model.
> Moreover, we have conducted a human evaluation in Table 5.
>
> **Learning discriminators for generation tasks & novelty**
>
> As the reviewer commented, our style transfer model uses the idea of learning discriminators to guide generation. Previous work in style transfer assumes distributionally matched corpora that differ only in the desired style. In this case, one can directly learn a discriminator between the two corpora and use it to guide the generation.
> Our task is a more challenging, in which the available data sources have additional confounding differences.
> For example, consider dialectal transfer, different dialects usually have a different set of speakers. If we directly learn a discriminator between two dialects, it will use not only dialectal features but also personal features of speakers in different dialects.
> There is no readily available data to learn a discriminator based solely on dialectal styles.
> We propose to infer the dialectal style as an invariant distinction across different speakers, and further introduce an orthogonal classifier to highlight speakers' personal styles. These two classifiers together can guide us to transfer the dialectal style without changing the speaker’s personal style.
> We have considered a novel setting that is closer to real-world style transfer and provided a novel solution.
>
> **Writing**
>
> We thank the reviewer for the suggestions. We will explain IRM in more detail and make it self-contained in the revision.
> In section 3.1.1, the invariant classifier will use features that are shared within each group, and our method requires that the provided datasets within each group share the desired style but not other features. We will clarify this in the revision.
>
> &nbsp;
>
> We hope that through the above discussions, the reviewer can appreciate the merits of this work. We will make the corresponding clarifications in the revision, and we are always available for further discussions.

---

### Official Review · Reviewer_9yNQ · 2021-11-04

**Correctness:** 3
**Technical Novelty And Significance:** 2
**Empirical Novelty And Significance:** 2
**Recommendation:** 3
**Confidence:** 4

**Main Review:**

The paper is tackling an important problem where we only want to change certain aspects of the text and maintains other characteristics.
The main concern is about the effectiveness of the proposed method. According to the human evaluation in Table 5, the other two baselines are doing better or close to the proposed method. I would also suggest adding human evaluation experiments on M w/C_ERM and M w / C_s. In Table 4, we also see that ERM loss achieves the best sentiment accuracy.  It shows better category accuracy, however, the category classifier only has a test accuracy around 71%, which makes the Category Acc column not very informative. The paper has demonstrated relatively good results on the synthetic sentiment transfer task. On the other hand, it would make the paper stronger if more real tasks and datasets can be evaluated.

Another missing part is the ablations studies of L_LM and L_BT in Eq 9. I understand that these two losses have been investigated in other works, it would still be good to know the benefits from them. I would also suggest adding qualitative results of ERM loss in Table 3 and Table 6, as well as the results of C_s in table 6. And a follow-up question is has the combination of C_ERM and C_o been tested ever? It might be interesting to take a look.

**Summary Of The Paper:**

This paper considers a text style transfer task where no paired sentences are available. For example, we have positive Clothing reviews and negative Cell Phone reviews, the paper tries to generate negative Clothing reviews. Note that it is likely the model transfers not only the sentiment (from positive to negative) but also product category (from clothing to cell phone). To avoid that, the paper proposes to learn an invariant style classifier (which can classify the sentiment regardless of the product category). Besides, the paper also proposes to learn an orthogonal classifier which can monitor any style-independent changes (e.g. product category).

**Summary Of The Review:**

The paper needs stronger empirical results. The proposed method is mainly built on top of Invariant Risk Minimization, thus lacks novelty. There are also ablation studies missing. I would recommend to reject this paper.

---

> ### Author Response · Authors · 2021-11-25
> **Response to reviewer 9yNQ**
>
> We thank the reviewer for the feedback. We address each question in turn:
>
> **Ablation studies**
>
> We thank the reviewer for raising the point. We have conducted experiments for the ablation studies, and the results are:
>
> | Synthetic task | Sentiment ACC | Punctuation ACC | PPL |	$\text{BLEU}_{\text{ref}}$ |
> | ----------- | ----------- | ----------- | ----------- | ----------- |
> | $C_s,C_o,\text{BT}$                                    |	96.1 | 49.6 | 2181.9	| 13.7 |
> | $C_s,C_o,\text{LM}$                                   |	81.0 | 90.3 | 4.9	| 0.7 |
> | $C_{\text{ERM}}, C_o,\text{LM},\text{BT}$ | 95.3 | 55.3 | 74.0     | 24.6 |
> | $C_s,C_o,\text{LM},\text{BT}$                    |  84.3 | 97.7 | 48.1    | 24.3 |
>
> | Real task | Sentiment ACC | Category ACC | PPL |	$\text{BLEU}_{\text{src}}$ |
> | ----------- | ----------- | ----------- | ----------- | ----------- |
> | $C_s,C_o,\text{BT}$                                    |	 77.4 | 49.0 | 1674.2 | 27.0 |
> | $C_s,C_o,\text{LM}$                                   |	99.5 | 18.0 | 5.4 | 1.9 |
> | $C_{\text{ERM}}, C_o,\text{LM},\text{BT}$ | 95.3 | 27.7 | 45.2 | 40.0 |
> | $C_s,C_o,\text{LM},\text{BT}$                    | 79.9 | 50.9 | 49.2 | 57.4 |
>
> Without the language model loss, the generated sentences have poor fluency (e.g., "the sales people here are terrible ." is transferred to "love sales people here are amazing ."), resulting in an extremely high perplexity.
> Without the back-translation loss, the model tends to generate generic outputs (e.g., different negative sentences are all transferred to "the food is great"), resulting in a very low BLEU score.
> Combining $C_{\text{ERM}}$ and $C_o$, we see that the model still does not preserve the confounding aspect well, as shown by its low Punctuation ACC and Category ACC. This is because unlike the invariant style classifier $C_s$, the ERM classifier makes predictions based on the confounding aspect and pushes the model to change it. Even if combined with $C_o$, this tendency has not been completely corrected.
>
> **Effectiveness of our method**
>
> The category classifier used for evaluation in Table 4 is 4-way, so the baseline accuracy is 25%. Although flawed, we think an accuracy of 71% still makes the classifier informative.
> According to the human evaluation in Table 5, Krishna et al. (2020) has content preservation close to ours, but its sentiment accuracy is worse than ours; He et al. (2020) has sentiment accuracy close to ours, but its content preservation is worse than ours.
> Overall, our model achieves a much higher success rate of transferring the sentiment while preserving the other aspects (18.7% higher than Krishna et al. (2020) and 6.2% higher than He et al. (2020)).
>
> **Novelty**
>
> Previous work in non-parallel style transfer assumes distributionally matched corpora that differ only in the desired style.
> We have considered a novel setting in which the available data sources are not distributionally matched, i.e., they differ not only in the desired style but also in confounding aspects.
> We show that in this challenging setting, it is possible to perform style transfer in the desired direction and generate types of sentences unseen during training.
> In particular, we propose to use subgroup annotations to illustrate the confounding difference, and make use of Invariant Risk Minimization to infer the desired style and the orthogonal aspects.
> We see our contribution as bringing the style transfer task closer to the real-world scenario, providing a methodology of what training signals to use and how to learn the model, and demonstrating for the first time that this more challenging problem can be solved.

---

### Decision · Program_Chairs · 2022-01-20

**Decision:**

Reject

**Comment:**

This paper studies text style transfer which aims to edit a given sentence to possess a desired style value (e.g., positive sentiment) while keeping all other styles and content unchanged. The paper specifically focuses on a challenging setting where besides the target style (e.g., sentiment) to transfer, there exists confounding attributes (e.g., product category) that correlate with the target style, making it hard to change only the target style while preserving the other. The proposed approach is to learn an invariant/unbiased style classifier using Invariance Risk Minimization (IRM), together with an orthogonal classifier for monitoring style-independent changes (e.g., product category), to supervise the generator training. The main concerns are on the experiments -- it's suggested to include experiments on other styles besides sentiment; human evaluation and/or other metrics are needed for more convincing comparison; it's also encouraged to experiment with large language models (e.g., GPT-2, BART) besides the small LSTM/CNN networks as in the present work.